



# Characterization of nucleosome sediments for protein interaction studies by solid-state NMR spectroscopy

**Ulric B. le Paige**[1], **ShengQi Xiang**[1,a], **Marco M. R. M. Hendrix**[2], **Yi Zhang**[3], **Gert E. Folkers**[1],
**Markus Weingarth**[1], **Alexandre M. J. J. Bonvin**[1], **Tatiana G. Kutateladze**[3], **Ilja K. Voets**[2], **Marc Baldus**[1],
**and Hugo van Ingen**[1]

[1]Utrecht NMR Group, Bijvoet Centre for Biomolecular Research, Utrecht University, 3584 CH,
Utrecht, the Netherlands
[2]Laboratory of Self-Organizing Soft Matter, Department of Chemical Engineering and Chemistry & Institute
for Complex Molecular Systems, Eindhoven University of Technology, P.O. Box 513, 5600 MB,
Eindhoven, the Netherlands
[3]Department of Pharmacology, University of Colorado School of Medicine, Aurora, CO 80045, USA
[a]current address: MOE Key Lab for Membrane-less Organelles & Cellular Dynamics, School of Life Sciences,
University of Science and Technology of China, 96 Jinzhai Road, Hefei, 230026, Anhui, China

**Correspondence:** Hugo van Ingen (h.vaningen@uu.nl)

**Abstract.** Regulation of DNA-templated processes such as gene transcription and DNA repair depend on the interaction of a wide range of proteins with the nucleosome, the fundamental building block of chromatin. Both solution and solid-state NMR spectroscopy have become an attractive approach to study the dynamics and interactions of nucleosomes, despite their high molecular weight of $\sim 200$ kDa. For solid-state NMR (ssNMR) studies, dilute solutions of nucleosomes are converted to a dense phase by sedimentation or precipitation. Since nucleosomes are known to self-associate, these dense phases may induce extensive interactions between nucleosomes, which could interfere with protein-binding studies. Here, we characterized the packing of nucleosomes in the dense phase created by sedimentation using NMR and small-angle X-ray scattering (SAXS) experiments. We found that nucleosome sediments are gels with variable degrees of solidity, have nucleosome concentration close to that found in crystals, and are stable for weeks under high-speed magic angle spinning (MAS). Furthermore, SAXS data recorded on recovered sediments indicate that there is no pronounced long-range ordering of nucleosomes in the sediment. Finally, we show that the sedimentation approach can also be used to study low-affinity protein interactions with the nucleosome. Together, our results give new insights into the sample characteristics of nucleosome sediments for ssNMR studies and illustrate the broad applicability of sedimentation-based NMR studies.

## 1 Introduction

Both prokaryotes and eukaryotes use an advanced protein machinery to regulate the expression and maintenance of their genome. Determining the molecular basis of the underlying interactions is crucial for our fundamental understanding of biology and for developing new treatments for disease. In prokaryotes, the regulatory proteins have direct access to the DNA. Ground-breaking NMR studies made a major contribution to our understanding of how such proteins search and recognize their target DNA sequences (Boelens et al., 1987; Spronk et al., 1999; Kalodimos et al., 2001, 2004). In eukaryotes, the DNA is packaged in nucleosomes, a protein–DNA complex formed by $\sim 145$–147 bp of DNA that are wrapped around a core of histone proteins (Fig. 1a). The histones H2A, H2B, H3 and H4 form an octameric complex that binds the DNA. The histones have N-terminal tails that are highly flexible and disordered, protruding from the

nucleosome core. Nucleosomes form an interaction platform for a multitude of proteins and protein complexes that regulate the function of chromatin (Fasci et al., 2018; Peng et al., 2020). Many of these bind to the histone proteins in the nucleosome, either to the histone tails or histone core, often depending on specific post-translational modifications of one of the histone proteins (McGinty and Tan, 2016; Speranzini et al., 2016). Nucleosomes can also be temporarily disassembled or moved as a consequence of protein interactions. Recent evidence indicates that these processes depend or at least involve internal dynamics of the histone proteins (Sanulli et al., 2019; Sinha et al., 2017).

Thanks to their unique sensitivity to molecular structure and dynamics, NMR studies have contributed greatly to our understanding of nucleosomes and nucleosome–protein complexes (see for a review van Emmerik and van Ingen, 2019). Thanks to the development of the methyl-TROSY approach (Tugarinov et al., 2003), it became possible to perform high-resolution NMR studies of histone protein interactions and dynamics within the nucleosome (Kato et al., 2011; Kitevski-LeBlanc et al., 2018). Following earlier work by the Jaroniec lab (Gao et al., 2013), our lab and the Nordenskiold lab recently introduced ssNMR-based methods to perform similar high-resolution studies on nucleosomes in a dense phase (Shi et al., 2018; Xiang et al., 2018). These approaches do not require selective isotope labeling of methyl groups as in methyl-TROSY solution NMR, thus offering to track interaction surfaces and histone protein dynamics along the full backbone. We refer the interested reader to a recent review detailing the pros and cons of the solution and solid-state-based approaches (le Paige and van Ingen, 2020). We used the ssNMR approach to determine the binding site of a high-affinity nucleosome-binding partner on the nucleosome core surface (Xiang et al., 2018). Shi, Nordenskiold and co-workers used ssNMR to determine internal histone dynamics in nucleosomes (Shi et al., 2018, 2020). Furthermore, similar studies are possible on nucleosomal arrays as models of native chromatin arrays, where multiple nucleosomes are assembled on a single, long DNA molecule (Shi et al., 2018).

In our approach (soluble) nucleosomes are sedimented using ultracentrifugation into an ssNMR rotor and then interrogated using $^1$H-detected ssNMR (Fig. 1b). This was inspired by seminal studies showing that sedimentation of soluble proteins results in high-quality samples for solid-state NMR, with the added benefit that sedimentation is fast, easy to use and does not perturb protein folding (Bertini et al., 2011; Fragai et al., 2013; Gardiennet et al., 2016; Mainz et al., 2015) and can be used to study protein–protein interactions (Bertini et al., 2013; Gardiennet et al., 2016). Recently, a thorough analysis showed that protein sediments are extremely stable, giving rise to highly reproducible ss-NMR spectra even years after rotor closure (Wiegand et al., 2020). Sedimentation has long been used to study the compaction of nucleosomal arrays (Osipova et al., 1980; Hansen et al., 1989). Nucleosomes are well-known to interact with

each other, mainly via interactions mediated by the histone tails (Garcia-Ramirez et al., 1992; Kan et al., 2007; Schwarz et al., 1996) in addition to other charge–charge interactions. As a result, nucleosome arrays can form various ladder-like or helical higher-order structures in vitro (Adhireksan et al., 2020; de Frutos et al., 2001; Garcia-Saez et al., 2018; Robinson et al., 2006; Schalch et al., 2005; Song et al., 2014), and this likely also underlies the observation of nucleosome clustering in vivo (Hsieh et al., 2015; Ricci et al., 2015). Recently, it was found that nucleosome arrays can also form condensates through liquid–liquid phase separation (Gibson et al., 2019). Notably, isolated nucleosomes also form tail-mediated interactions with one another – so-called in trans interactions (Bilokapic et al., 2018) – and isolated nucleosomes are able to stack into columns in highly concentrated solutions, as shown schematically in Fig. 1c (Berezhnoy et al., 2016; Bertin et al., 2007a; Leforestier and Livolant, 1997; Livolant et al., 2006; Mangenot et al., 2003a, b). This strong propensity for in trans interactions could potentially be favored in the high concentration samples obtained through the sedimentation approach used in our NMR studies. The formation of high-order structures in which specific nucleosome surfaces are involved in inter-nucleosome interactions could both alter their intrinsic internal dynamics and reduce their availability for protein interactions.

Here, we examined the packing of nucleosomes in the sediment and explored its impact on nucleosome–protein interaction studies. Through careful sample analysis, we found that the nucleosome concentration in the sediment is $\sim 2.4$ mmol/dm$^3$ with a packing ratio of $\sim 55\%$–60 %. The sediments are devoid of pronounced long-range ordering of nucleosomes according to SAXS experiments, indicating that inter-nucleosome interactions within the sediment are highly heterogenous and likely dynamic in nature. To assess the impact on the study of nucleosome–protein interactions, we focused on the second PHD finger of CHD4 as a test case. This protein binds weakly to the histone H3 tail (Musselman et al., 2009), which is one of the main inter-nucleosome contact sites (Gordon et al., 2005; Kan et al., 2007) and must thus compete with the nucleosomal DNA in order to bind (Gatchalian et al., 2017). Upon addition of PHD2, we observed highly similar effects in both solution and solid-state H3 NMR spectra, indicating that the sedimentation approach can in principle also be applied for the many proteins that bind nucleosomes with low affinities and/or through the highly flexible histone tails. Together, our results give new insights into the sample characteristics of nucleosome sediments for ssNMR studies and illustrate the broad applicability of sedimentation-based NMR studies.

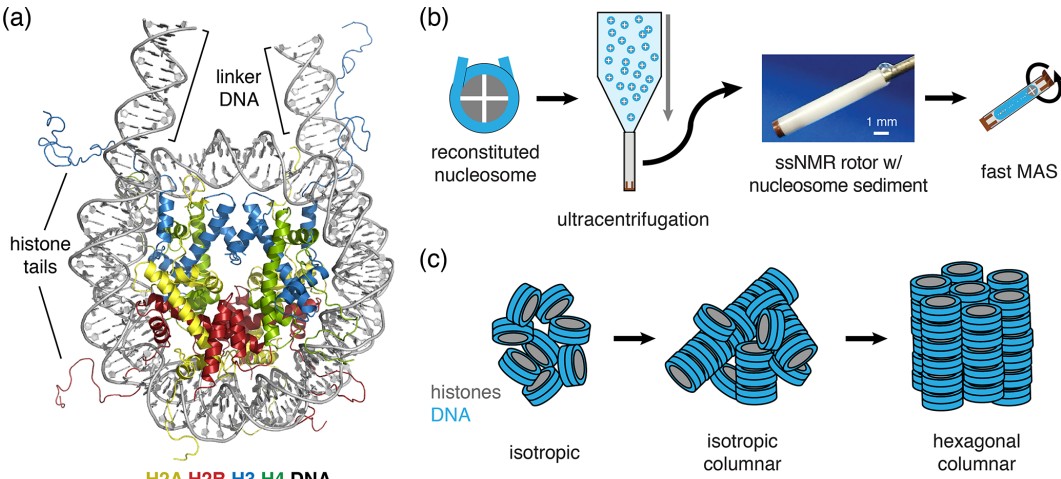

**Figure 1.** Schematic of nucleosome structure and sedimentation-based nucleosome NMR studies. **(a)** Structure of the nucleosome based on the crystal structure of the nucleosome core particle, extended with 10 bp of linker DNA at each end. Linker DNA and two of the N-terminal histone tails (of one H3 and one H2B copy in the nucleosome) are indicated. Color coding indicated in the figure. **(b)** Overview of the sedimentation-based ssNMR study of nucleosomes. A dilute solution of nucleosomes is ultracentrifuged directly into the 1.3 mm rotor to create a nucleosome sedimentation for $^1$H detected ssNMR studies. A droplet of viscous liquid is visible at the top of rotor. **(c)** Schematic of nucleosome packing in dense phase as (from left to right) an unordered isotropic, isotropic columnar or highly ordered hexagonal columnar stacking of nucleosomes.

## 2  Materials and methods

### 2.1  Sample preparation

Three nucleosome samples that are further characterized in this study were prepared previously and described in Xiang et al. (2018). These nucleosome samples contain, respectively, isotope-labeled H2A, H3 or H2A with co-sedimented LANA peptide and are listed as samples 1–3 in Table 1 below. Isotope-labeled histones were fractionally deuterated to reduce line width and increase sensitivity in $^1$H-detected ss-NMR experiments (Mance et al., 2015). For this study we prepared two new H3-labeled nucleosome samples, one with nucleosomes in their free state (sample 4 in Table 1) and with a co-sedimented PHD2 domain of CHD4 (PHD2). We additionally prepared one natural abundance nucleosome sample exclusively for the solution SAXS experiment. All were prepared as described in Xiang et al. (2018). Briefly, recombinant *Drosophila melanogaster* histones were expressed as inclusion bodies in *E. coli* BL21(DE3) Rosetta2 grown in either lysogeny broth (LB) for unlabeled histones or deuterated M9 with $^1$H, $^{13}$C-glucose and $^{15}$NH$_4$Cl (used both for solution NMR and $^1$H-detected ssNMR). The cells were lysed with a French press, and inclusion bodies were washed with triton X-100, solubilized in guanidine chloride and purified in urea by gel filtration and ion exchange chromatography. Pure histones were mixed equimolarly and dialyzed to high salt into histone octamers, which was purified by gel filtration. A pUC19 plasmid harboring 12 copies of a 167 bp version of the 601 DNA sequence (Lowary and Widom, 1998) was amplified in *E. coli* DH5α and purified by alkaline lysis

and ion exchange chromatography. The plasmid was then restricted with Sca1 and the 601 DNA fragment was purified by ion exchange chromatography. Histone octamers and DNA were mixed at 1 : 1.04 molar ratio in high salt and gradient-dialyzed to low salt. The reconstituted nucleosomes were dialyzed to PK10 buffer (10 mmol/dm$^3$ potassium phosphate supplemented with 10 mmol/dm$^3$ KCl, pH 6.5), reconstitution efficiency was checked by native PAGE and concentration checked by UV absorbance at 260 nm using the DNA sequence-specific absorbance coefficient and the individual predicted histone molar extinction coefficients at 260 nm, calculated as $\varepsilon_{260} = \varepsilon_{280} \times 0.54$ (see Dyer et al., 2004; Xiang et al., 2018). The PHD2 finger domain from CHD4 was produced as described in Musselman et al. (2009). In brief, CHD4 PHD2 (443–498) was expressed in *Escherichia coli* BL21 DE3 pLysS cells grown in LB media. Protein expression was induced with 0.5 ∼ 1 mmol/dm$^3$ IPTG for 16 h at 16 °C. The GST-tagged protein was purified on glutathione Sepharose 4B beads (GE Healthcare) in 20 mmol/dm$^3$ Tris-HCl (pH 6.8) buffer, supplemented with 150 mmol/dm$^3$ NaCl and 3 mmol/dm$^3$ DTT. The GST tag was cleaved overnight at 4 °C with PreScission or Thrombin protease. The cleaved PHD2 protein was further purified by size exclusion chromatography and buffer exchanged into the low-salt PK10 buffer prior to lyophilization for storage. For preparing the NMR samples, CHD4-PHD2 dialyzed to either low-salt PK10 buffer or high-salt PK buffer with 100 mmol/dm$^3$ KCl (PK100).

**Table 1.** Estimated nucleosome concentration in sediment.

| Sample id | Sample 1 | Sample 2 | Sample 3 | Sample 4 |
|---|---|---|---|---|
| Sample type | H3-labeled | H2A-labeled | H2A-labeled + LANA | H3-labeled |
| Nucleosome mass[a] (mg) in | | | | |
| – initial starting solution | 1.98 | 1.90 | 1.90 | 2.76 |
| – supernatant after sedimentation | 0.06 | 0.09[b] | 0.11 | 1.03 |
| – cap clearing volume | 0.38[c] | 0.35[c] | 0.35[c] | 0.11[d] |
| – rotor | 1.54 | 1.45 | 1.44 | 1.59 |
| Final nucleosome concentration in rotor[e]: | | | | |
| In mg/cm$^3$ | 514 | 484[b] | 481 | 529 |
| In mmol/dm$^3$ | 2.43 | 2.29 | 2.28 | 2.50 |

[a] Based on absorbance measurements at 260 nm using the DNA sequence-specific absorbance coefficient and the individual histone-predicted molar extinction coefficients at 260 nm, calculated as $\varepsilon_{260} = \varepsilon_{280} \times 0.54$.
[b] Assuming 95 % sedimentation efficiency.
[c] Assuming a homogenous nucleosome distribution in the rotor and cleared space volume of 0.73 μL.
[d] Measured by diluting the cleared material in buffer and measuring absorbance.
[e] Calculated using an internal volume of 3 μL for the 1.3 mm rotor.

## 2.2   Solution-state NMR experiments

Solution-state NMR experiments for the interaction study of PHD2 and the nucleosome were performed on a Bruker 21.1 T magnet equipped with an Avance III console and a CPTCI probe, at a temperature of 298 K. NMR samples contained ~ 36 μmol/dm$^3$ nucleosome with fractionally deuterated, $^{13}$C,$^{15}$N-labeled H3 in PK10 buffer with 10 % of $D_2O$, 0.01 % NaN$_3$ and protease inhibitors. PHD2 in either PK10 or PK100 buffer was titrated to this sample and chemical shift and peak intensity changes were monitored using 2D $^{15}$N–$^1$H TROSY HSQC spectra ($t_{1,\max}$ 122 ms, $t_{2,\max}$ 67 ms, total acquisition time per spectrum ~ 2 h). The FID was apodized in both dimensions with a squared cosine bell function and extended once by linear prediction in the indirect dimension before Fourier transform. Free nucleosome spectra were recorded in both low-salt PK10 buffer and high-salt PK100 buffer.

## 2.3   Solid-state NMR experiments

Sedimentation of samples for $^1$H-detected ssNMR studies was carried out as described in Xiang et al. (2018). Briefly, a custom-made filling device as described in Narasimhan et al. (2021), though other similar designs exist, see for example Bertini et al. (2012), Böckmann et al. (2009) and Mandal et al. (2017), loaded with a 1.3 mm Zirconia rotor (Bruker) was filled with a solution containing ~ 2 mg nucleosome with fractionally deuterated, $^{13}$C, $^{15}$N-labeled histone in PK10 buffer. For co-sedimentation of PHD2, nucleosome and PHD2 were mixed in a 1 : 40 molar ratio (corresponding to a 20 : 1 molar ratio to H3 tail) in PK100 buffer and incubated for 10 min. Subsequently, MgCl$_2$ was added from a 4 mmol/dm$^3$ stock solution in PK10 or PK100 buffer to 2 mmol/dm$^3$ Mg$^{2+}$. The filling device was loaded in an ultra-

centrifuge (Beckman-Boulter Optima L-90K) with a swinging bucket SW 32 TI rotor and centrifuged at 83 000 g for 24–28 h at 4 °C. After removal of the supernatant, the rotor was recovered and the top cleared before closing the rotor by placing the cap without further sealing or inserts.

Solid-state NMR experiments were performed in a Bruker 18.8 T magnet equipped with 1.3 mm $^1$H/X/Y triple-resonance MAS probe spinning at 50 kHz MAS at ~ 298 K TS1. The 2D J-based and CP-based $^1$H-detected NH spectra were recorded as described in Xiang et al. (2018). The J-based NH spectrum was acquired with $t_{1,\max}$ 20 ms, $t_{2,\max}$ 20 ms and a total acquisition time of ~ 5 h. The FID was apodized with a 30° shifted squared cosine bell function in both dimensions and zero-filled twice in both dimensions, and indirect dimension was extended once by linear prediction before Fourier transform. The CP-based NH was acquired with $t_{1,\max}$ 21 ms, $t_{2,\max}$ 20 ms and a total acquisition time of ~ 10 h. The FID was apodized with an exponential window function with line broadening of 50 Hz in the direct dimension and a 30° shifted squared cosine bell function in the indirect dimension, both dimensions were zero-filled twice and the indirect dimension was extended by linear prediction before Fourier transform.

## 2.4   NMR data analysis

All NMR data were processed in Bruker Topspin and analyzed in NMRFAM-Sparky (Lee et al., 2015). Assignments of the histone H2A and H3 tail resonances were taken from Xiang et al. (2018). Chemical shift perturbations (CSPs) were calculated as the 2D peak displacement in ppm using a weighting factor of the $^{15}$N chemical shift differences (in ppm) of 0.154 (Williamson, 2013). For the calculation of peak intensity ratios, peak intensities in individual spectra

were scaled by the number of scans, receiver gain setting, Bruker nc_proc parameter and, for solution NMR experiments, dilution factor. Errors in peak intensities were based on the spectral noise level.

## 2.5 SAXS experiments

A solution of $6 \, \mu mol/dm^3$ (approximatively $1.3 \, mg/cm^3$) nucleosome in PK buffer, and the nucleosome sediments in open air, were loaded in $2 \, mm$ quartz capillaries (Hilgenberg GMBH) sealed with wax. The SAXS measurements were carried out on a SAXSLAB GANESHA 300 XL system equipped with a GeniX 3D Cu Ultra Low Divergence micro focus sealed tube source producing X-rays with a wavelength $\lambda = 1.54 \, \text{Å}$ at a flux of $1 \times 10^8 \, ph/s$ and a Pilatus 300K silicon pixel detector with $487 \times 619$ pixels of $172 \, \mu m \times 172 \, \mu m$ in size. The beam center and $q$ range were calibrated using silver behenate as a standard. Two sample-to-detector distances were used of 713 and 1513 mm, respectively, to access a $q$ range of $0.06 \leq q \leq 0.44 \, \text{Å}^{-1}$ with $q = 4\pi/\lambda$ $(\sin \theta / 2)$. Each profile recorded at 713 and 1513 mm comprises 960 successive captures with 15 s pause. Medium- and small-angle data were merged. Data analysis was made using the PRIMUS and GNOM programs from the ATSAS v3.03 suite (Manalastas-Cantos et al., 2021). Backgrounds were PK buffer and an empty section of the capillary for soluble nucleosome and sediment samples, respectively. Points within 0.007 and $0.03 \, \text{Å}^{-1}$ and within 0.007 and $0.186 \, \text{Å}^{-1}$ were used for the molecular weight analysis and the determination of the distance distribution function, respectively.

## 2.6 Modeling of the PHD2–nucleosome complex

The PHD2 domain of CHD4 (extracted from PDB entry 2LZ5) was docked to one of the two H3 tails in the nucleosome using the HADDOCK 2.4 webserver (van Zundert et al., 2016). As input structure, we used a molecular model for our experimental system of a nucleosome containing Dm. histones and 167 bp of 601-DNA. This model was based on the crystal structure of the nucleosome from *Xl.* histones and 147 bp of alpha-satellite DNA (PDB entry 1KX5). The histone sequences were mutated using Modeller (Webb and Sali, 2016), the DNA sequence mutated and extended with 10 bp of B-form DNA at each end using the 3D-DART webserver (van Dijk and Bonvin, 2009). Docking was guided by unambiguous interaction restraints derived from the complex structure of the PHD2 domain with a H3 tail peptide (PDB entry 2LZ5). The H3 tail residues 1–8 in the nucleosome were defined as fully flexible segments for the docking. Otherwise default docking parameters were used. The final 200 solutions clustered into a single cluster. To investigate potential DNA binding by PHD2, ambiguous interaction restraints were defined between R94, K97, R133, K140, K142 and the 1.5 turn of DNA surrounding the H3 tail exit site. The H3 tail residues 1–27 were defined as fully flexible, and to allow larger conformational changes, the number of MD steps were increased to 2000/2000/4000/4000 for the various stages of the flexible refinement stage (a factor 4 increase compared to the default) as described for protein–peptide docking (Trellet et al., 2013). In this case, the final 200 solutions clustered into four clusters. The largest but not top-scoring cluster (147 members) did not show any PHD2–DNA contacts. The best-scoring cluster (26 members) showed consistent PHD2–DNA contacts while maintaining the native H3 tail interaction mode. The four best solutions of the best-scoring cluster were analyzed using PyMOL (Schrödinger, LLC, 2015).

## 3 Results

### 3.1 Nucleosomes are tightly packed in the sediment.

As a first characterization of the nucleosome sediment in the ssNMR rotor, we assessed the nucleosome concentration for four different sample preparations from absorbance measurements of the solution before and after ultracentrifugation. Three of the four samples analyzed were prepared as part of our initial study (Xiang et al., 2018) and one as part of an ongoing investigation. In all cases, the sedimentation process was started from a $0.5 \, cm^3$ solution containing $4 \, mg/cm^3$ ($\sim 20 \, \mu mol/dm^3$) nucleosomes (with or without a binding partner), placed in a custom-made device. This is then centrifuged at $83\,000 \, g$ into a $1.3 \, mm$ ssNMR rotor. As can be seen from Table 1, the homogenized supernatant after sedimentation retains, with one exception, only $2 \, \% – 5 \, \%$ of the initial UV absorbance, indicating a near-quantitative sedimentation. The efficiency of sedimentation roughly matches that predicted using the sedNMR webtool (Ferella et al., 2013), when considering that the favorable inter-particle interactions between nucleosomes may lower the threshold for immobilization. For sample 4 a much higher nucleosome concentration in the supernatant was observed, but this can be rationalized by the also much higher starting mass. Upon removal of the sediment from the very top of the rotor to make room for placement of the rotor cap, a transparent, viscous droplet was formed in all cases (see Fig. 1b). This indicates that the rotor is filled with a dense solution rather than a precipitate. The final nucleosome mass in the rotor is estimated to be 1.44–1.59 mg, resulting in concentrations in the range of 480 to $530 \, mg/cm^3$ or 2.3 to $2.5 \, mmol/dm^3$. This value is similar to the in-rotor concentration reported by Shi et al. (2018) using $Mg^{2+}$-induced precipitation of nucleosomes. Notably, for sample 4 a much higher nucleosome mass was used in the sedimentation mix compared to samples 1–3 ($\sim 45 \, \%$ more). This resulted in only a $\sim 5 \, \% – 10 \, \%$ increase in final nucleosome concentration, indicating the nucleosome packing is close to maximum at this centrifugation speed. For comparison, the local maximum concentration of nucleosomes in the cell is estimated to range between 0.25 and $0.5 \, mmol/dm^3$ (Nozaki et al., 2013; Weidemann et al., 2003). Assuming the nucleo-

somes to be homogeneously distributed through the volume of the packed rotor and approximating the nucleosome to a disk-shaped object of 420 nm$^3$ (van Vugt et al., 2009), the observed nucleosome concentration corresponds to a packing ratio of $\sim$ 55 %–60 %. These concentrations and packing ratios of the nucleosome sediment are lower than those found in nucleosome crystals. Based on crystallography parameters from four nucleosome crystal structures (Protein Data Bank (PDB) entries 2PYO, 1KX5, 1AOI and 3LZ0, Clapier et al., 2008; Davey et al., 2002; Luger et al., 1997; Vasudevan et al., 2010), we find that the typical concentrations are $\sim$ 3.2 mmol/dm$^3$, corresponding to a particle packing coefficient of $\sim$ 76 %. These considerations indicate that the sediment is highly dense with a packing ratio close to 80 % of that in crystals, suggesting that a significant amount of ordering and nucleosome–nucleosome interactions may occur.

## 3.2   Nucleosomes are stably folded and remain hydrated in the sediment during NMR measurements.

We previously reported $^1$H-detected ssNMR spectra of sedimented nucleosomes containing either isotope-labeled histone H2A or H3 (Xiang et al., 2018). The backbone chemical shifts together with the high quality of the spectra indicated that the histone proteins were folded as in the nucleosome crystal structure. We here re-examined the spectra obtained on these samples to assess sample hydration and histone folding over time and to check for signs of inter-nucleosome interactions.

The 1D single-pulse $^1$H NMR spectrum of the sediment is dominated by an intense water signal, indicating the nucleosome sediment is highly hydrated. Comparison of these spectra throughout the measurements for the H2A-labeled nucleosome shows that the water signal remains prominent over time, despite exposure to 34 d of high-speed MAS at an effective temperature of 37 °C. The intensity at peak maximum decreases by 20 % over this time, while the line width increases by 40 % (Fig. 2a). Similar results were obtained for other samples. A minor component of free bulk water can be observed in the earliest spectra (Böckmann et al., 2009), which disappears over time due to evaporation. We conclude that, while some degree of dehydration occurs, the sediment samples remain overall well-hydrated throughout the NMR measurements.

To assess histone fold over time, we compared 2D cross-polarization (CP)-based NH correlation spectra recorded at the beginning and the end of the measurements, across a 5-month period. Both spectra are of high quality, showing a well-resolved and well-dispersed spectrum (Fig. 2b). There are little chemical shift or intensity changes between the spectra, indicating the histones remain well-folded over time. The slight chemical shift changes (less than the line width) are observed for few H2A residues, most of which are in the vicinity of buried waters or salt ions in the crystal structure (Clapier et al., 2008; Materese et al., 2009). This could be re-

lated to changes in the hydration as seen from the 1D spectra. For H3, no differences in peak positions over time could be resolved (data not shown).

Since the nucleosome concentration in the sediment is ca. 25–50-fold higher than that in typical solution NMR samples, comparison of solid-state and solution NMR spectra may reveal insights into inter-nucleosome interactions. We previously reported that J-based ssNMR spectra of H2A- or H3-labeled nucleosomes have a highly similar chemical shift to solution, indicative of fast tail motion in the sediment. Within nucleosome arrays, the histone tails have been shown to be involved in inter-nucleosome interactions, while in single nucleosomes they bind the nearby DNA within the same nucleosome (Stützer et al., 2016; Shaytan et al., 2016). The close chemical shift correspondence between the ssNMR and solution spectra could thus mean that within the sediment the histone tails bind to DNA within the same nucleosome, as in dilute solution. However, given the dense packing of nucleosomes, this is rather improbable. Rather, the observed chemical shifts likely do not permit us to discriminate whether the histone tail–DNA interaction occurs in an intra- or inter-nucleosomal fashion.

In addition to the non-specific histone tail–DNA interactions, a specific interaction between the H4 tail and the H2A surface mediates nucleosome–nucleosome contacts that are required for compaction of chromatin fibers (Kalashnikova et al., 2013). The backbone chemical shifts of a H2A dimer within the sediment can only be compared to solution chemical shifts of a H2A–H2B dimer, due to the molecular weight limit for amide-based solution NMR. This comparison revealed no significant chemical shift differences for the H2A residues that are involved in H4 tail binding, indicating that there is no stable inter-nucleosome interaction within the sediment. Taken together these data indicate the nucleosomes in the sediment remain well-folded and hydrated through the measurements without evidence for direct nucleosome–nucleosome contacts.

## 3.3   Nucleosome sediments are 3D networked gels lacking long-range ordering

To allow further investigations, we recovered the contents of the ssNMR rotor for the H2A-labeled nucleosome (sample 2 in Table 1, spectra shown in Fig. 2), the H2A-labeled nucleosome bound to the LANA peptide (sample 3 in Table 1) and the H3-labeled nucleosome (sample 4 in Table 1). The recovered sediments appeared as transparent semi-solid gels. One sample (sample 4) was highly viscous, whereas two others (samples 2 and 3) had a rather paste-like solidity (Fig. 3a). Part of this "nucleosome paste" was resuspended for native PAGE analysis, confirming that the nucleosomes had remained intact throughout the measurements and storage period (Fig. 3b). There was no correlation between the observed solidity and obvious experimental conditions such

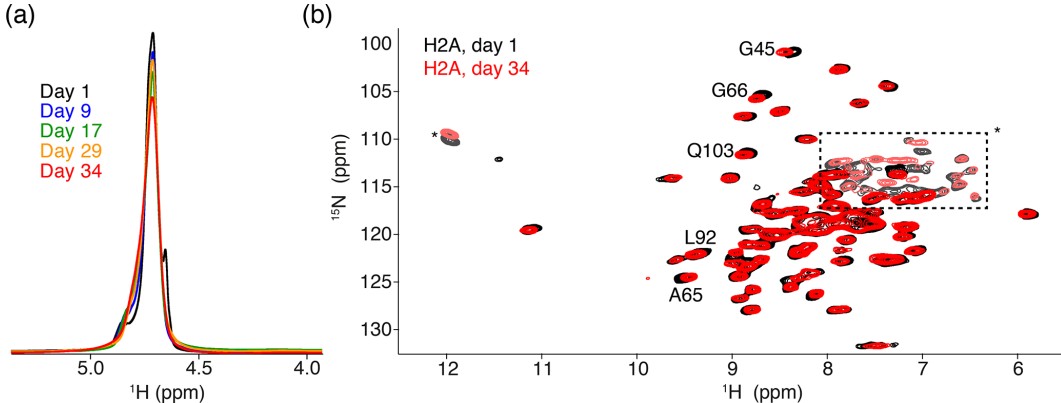

**Figure 2.** Comparison of NMR data recorded directly after sedimentation and after 5 months. **(a)** Overlay of the 1D one-pulse $^1$H spectrum showing the highly dominant water signal. Spectra are annotated with the cumulative number of days of ssNMR measurements (total of 34 d). The sample was stored in between measurement sessions at 4 °C. **(b)** Overlay of the 2D $^1$H-detected CP-based NH correlation spectrum acquired at the beginning and end of the NMR measurements. Resonances with slight chemical shift changes are indicated. Resonances with light color, indicated by an * or in the dashed box, are from side-chain resonances. Some of these side-chain resonances are folded into a different position along the $^{15}$N dimension due to use of a different offset frequency. Color coding for both panels indicated in the figure.

as nucleosome concentration, NMR measurement time, or sample age.

Gelation is a well-known property of polymers that can create a 3D meshwork through covalent or non-covalent interactions. Thus, the observation of gel-like material properties for the nucleosome sediment conclusively demonstrates the presence of significant inter-nucleosome interactions. While the semi-solid appearance of the sediment may at first sight suggest significant dehydration, its transparency rather suggests the sediment is a hydrogel that retains significant amounts of water. We speculate that the gradual increase in water line width may correlate with the transition to a semi-solid hydrogel.

To investigate the packing and ordering of nucleosomes in the recovered sediments, we turned to SAXS experiments. First, SAXS data collected on a nucleosome solution resulted in a scattering curve consistent with monodisperse particles with a radius of gyration of 5.7 nm and maximum extension of 13.3 nm (Fig. 3c, Table 2). These values match well to the radius and end-to-end length of a nucleosome with 10 bp of linker DNA, respectively. Also, molecular weight estimated from the SAXS data (208 kDa; see Table 2) agrees well with the expected mass of 211.3 kDa. As expected, the recovered sediments show a strikingly different scattering profile (Fig. 3d). While each sample showed overall somewhat different scattering curves, all featured a pronounced peak at $q^* \sim 0.08$, corresponding to a characteristic distance of $\sim 7$–8 nm. For the H2A-labeled nucleosome "paste" (sample 2; black curve) a second broad peak was observed at $q^* \sim 0.16$, suggestive of a laminar organization with a main characteristic distance of $\sim 7$ nm. The very broad appearance of the scattering peaks either reflects a heterogeneous distribution of the characteristic distance across the sample or indicates that the organization is only regular over a short distance. In samples 3 (purple curve) and 4 (blue curve), the first reflection at $q^* \sim 0.08$ features also a relatively sharp component, suggesting that in these samples there is a more structured subpopulation.

While we observed sample-to-sample variation, the sediments seem to primarily consist of heterogeneously packed nucleosomes with mean inter-particle distances of 7–8 nm. While some short length structures cannot be excluded, the SAXS measurements demonstrate that the nucleosome sediments are devoid of pervasive long-range ordering.

### 3.4 Co-sedimentation of a weak, histone tail-binding protein

Having established that the nucleosome sediment in our studies is not strongly ordered and is thus likely to only minimally interfere with protein binding, we next sought to stringently test the co-sedimentation approach. While we previously con-sedimented a peptide that binds with very high affinity to the histone core surface, we here used a protein domain that weakly binds to the histone H3 tail, the second PHD finger (PHD2 hereafter) of CHD4. This chromatin remodeler protein is part of the NuRD complex that is involved in DNA repair and cell cycle progression (Allen et al., 2013). Recruitment of CHD4 to chromatin depends on the interaction of its paired PHD finger domains (PHD1 and PHD2) with the H3 tail (Gatchalian et al., 2017; Mansfield et al., 2011; Musselman et al., 2012). Both PHD1 and PHD2 bind non-modified H3 tail peptides with micromolar-range affinity (Mansfield et al., 2011; Musselman et al., 2009). However, solution NMR titration experiments with nucleosomes showed that binding of PHD2 to the nucleosome is reduced compared to the binding of PHD2 to histone H3 peptides, indicating a pronounced inhibitory effect of the nucleosomal

**Table 2.** Analysis of SAXS data of soluble nucleosomes.

| Guinier analysis – DATMW | Mononucleosomes, 167 bp 601 DNA |
| --- | --- |
| I(0) (cm$^{-1}$) | $0.089 \pm 3.8 \times 10^{-5}$ |
| $R_g$ (Å) | $42.86 \pm 0.03$ |
| $q_{min}$ (Å$^{-1}$) | 0.0007 |
| q$R_g$ max ($q_{min} = 0.0007$ Å$^{-1}$) | 1.3 |
| Bayesian inference | |
| MW estimate (Da) | 208 000 (83.56 % probability) |
| Credibility interval (99.75 % probability) | [176 600, 221 050] |
| $P(r)$ analysis (GNOM) | |
| $I(0)$ (cm$^{-1}$) | $0.089 \pm 3.8 \times 10^{-5}$ |
| Rg (Å$^{-1}$) | $44.13 \pm 0.09$ |
| $d_{max}$ (Å) | 133 |
| $Q$ range (Å$^{-1}$) | 0.007–0.187 |
| $\chi^2$ (estimate from GNOM) | 0.85 |
| Porod volume (Å$^{-3}$) | 356 874 |

environment (Gatchalian et al., 2017). At least part of the reduced binding affinity can be explained by the reduced availability of the H3 tail for binding within the nucleosome, as a result of DNA binding by the H3 tail (Stützer et al., 2016). We here investigated whether PHD2 can overcome the competition effect from the DNA and bind the H3 tail within the sediment (Morrison et al., 2018). By observing the nucleosome rather than the PHD2 domain, the nucleosome sample requirements can be reduced, allowing the investigation of such weak interactions.

As a control experiment, we first assessed binding of PHD2 to nucleosomes by solution NMR. Titrating unlabeled PHD2 to H3-labeled nucleosomes to a 2 : 1 molar ratio at low salt (25 mmol/dm$^3$ ionic strength, PK10 buffer) did not result in significant spectral changes (data not shown). At high salt (125 mmol/dm$^3$ ionic strength, PK100 buffer), however, PHD2 binding was visible as a peak intensity decrease for residues in the H3 tail (Fig. 4a, b). Residues T3, K4, T6 and A7 showed the largest intensity reduction, which, when fitted to a single binding site model, yielded a $K_D$ of $168 \pm 8$ mol/dm$^3$ TS2. Notably, no significant chemical shift perturbations or new signals were observed, even after addition of 10 molar equivalents PHD2 to H3 tail, suggesting the bound state of the H3 tail is invisible in solution NMR.

We next co-sedimented unlabeled PHD2 and H3-labeled nucleosomes by ultracentrifugation of a solution containing PHD2 and nucleosome at molar ratio PHD2 : H3 tail 20 : 1 at high salt (PK100 buffer). The supernatant after sedimentation contained mostly PHD2 according to absorbance measurements and gel analysis, suggesting that the specific complex was sedimented while the excess PHD2 remained in solution. Again, a viscous droplet was recovered while clearing space for rotor closure. Both J- and CP-based NH spectra were recorded at 50 kHz MAS. Both spectra were of high

quality with well-resolved and well-dispersed resonances (Fig. 4c, d). Comparison to spectra of free, sedimented H3-labeled nucleosome revealed no resolvable changes in peak position. Notably, the peak intensity profiles in the J-based spectrum, probing the flexible parts of H3, indicate a similar residue-specific drop in peak intensity as observed in solution (Fig. 4e). Even if the lack of resolution in these spectra hinders interpretation somewhat, it can clearly be seen that resonances for the first 10 tail resides show significantly decreased peak intensities, down to 30 %–50 % of the original intensity. Again, no new peaks corresponding to the bound state could be observed, suggesting rigidification of the H3 tail in the bound state. Careful examination of the CP spectra unfortunately also did not reveal any new resonances, suggesting that the bound state is not fully rigid but most likely exhibits dynamics on a timescale faster than milliseconds.

We used spectra of H3-labeled nucleosome sediment recorded at 25 mmol/dm$^3$ ionic strength (PK10 buffer) as a reference, as spectra of sedimented, free nucleosomes at 125 mmol/dm$^3$ ionic strength (PK100 buffer) were not available. To rule out the possibility that the increased salt concentration caused reduced intensity of terminal H3 tail residues, we compared solution NMR spectra recorded at 25 and 125 mmol/dm$^3$ ionic strength. Addition of salt resulted in small chemical shift perturbations for several residues in the stretch 19–29, signifying a slight shift from a DNA-bound to DNA-free state (Stützer et al., 2016). These chemical shift changes are too small to be resolved in the ssNMR spectra. Furthermore, addition of salt approximately doubled the peak intensity for many residues in the 19–29 region, indicating increased flexibility for this part of the H3 tail (Fig. 4f) and explaining the higher relative peak intensities for this stretch in Fig. 4e. Importantly, no peak intensity changes in the PHD2 binding site could be discerned.

Magn. Reson., 2, 1–16, 2021 https://doi.org/10.5194/mr-2-1-2021

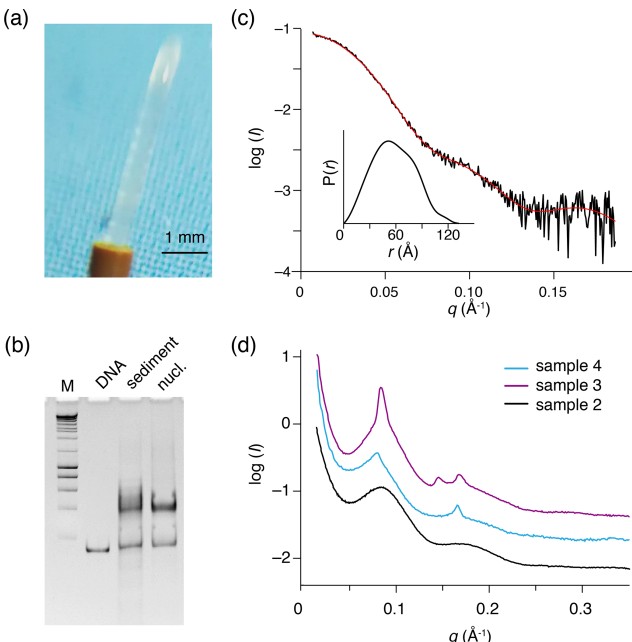

**Figure 3.** Recovered nucleosome sediment and SAXS scattering curves. **(a)** The nucleosome sediment of sample 2 (H2A-labeled nucleosomes) recovered from the ssNMR rotor after 34 d of MAS and 11 months of storage at 4 °C appears as a transparent semi-solid, paste-like gel. **(b)** Native PAGE analysis of the recovered sediment (sample 4, H3-labeled nucleosomes, lane 3) together with free 167 bp DNA (lane 2) and a fresh reconstituted nucleosome (lane 3). DNA base-pair marker in lane 1. Positions of free DNA and nucleosomes are indicated. Presence of a pronounced nucleosome band with little free DNA indicates the recovered sediment consists of nucleosomes. **(c)** SAXS-based scattering curve of nucleosomes in solution (6 µmol/dm$^3$, 1.3 mg/cm$^3$) in PK10 buffer. The buffer-subtracted scattering profile (black) was fitted in GNOM as a monodisperse particle function (red). Inset shows the derived pair distance distribution function. **(d)** SAXS-based scattering curves of the recovered nucleosome sediments, color coding indicated. All three samples feature a distinctive peak at $q^* \sim 0.08$, corresponding to a characteristic distance of 7.8 nm ($d = \frac{2\pi}{q}$). The scattering curve of sample 3 (H2A-labeled nucleosome with co-sedimented LANA) features few relatively sharp peaks, indicative of more long-range ordering. Notably, this sample was the least solid-like.

We conclude that the PHD2 finger can be co-sedimented with the nucleosome despite the low binding affinity and that specific binding of the PHD2 finger to the H3 tail can be demonstrated using the sediment ssNMR approach. Unfor-
5 tunately, the PHD2-bound state is not directly observable, preventing further detailed structural characterization of the bound H3-tail conformation.

## 4 Discussion

We here characterized in some detail the nucleosome sed-
10 iment that is central to our ssNMR investigation of nucle-

osome dynamics and nucleosome–protein interactions. We find that the sedimentation procedure is robust and reproducible. The nucleosome concentration in the sediment approaches that observed in a crystal. Nucleosomes remain well-folded and, despite some water loss, remain hydrated in 15 the sediment over the course of several weeks of MAS. The recovered sediments appear as translucent gels with semi-solid properties, which lack strong long-range ordering based on SAXS measurements. The sediment thus likely corresponds to a dense network of nucleosomes with transient 20 and continuously rearranging inter-nucleosome interactions (Fig. 5). Judging from the large width of the first peak, we can roughly estimate that the length scale of the regular structure in the sediment is ∼ 15–20 nm, corresponding to stacks of two to three nucleosomes, at least for sample 2. 25

The interactions between nucleosomes are mediated by the histone tails (Bendandi et al., 2020; Kan et al., 2007; Stützer et al., 2016), possibly together with other stabilizing contacts (Bilokapic et al., 2018). As a result the nucleosomes are packed close enough to prevent overall tumbling but dis- 30 tant enough to allow continuous rearrangement, preventing long-range ordering. This view is consistent with homogeneous chemical environment of the histone spins as seen from NMR and the heterogenous ordering on a macroscopic scale as seen from SAXS. 35

The high spectral quality and long-term stability of sedimented proteins and protein-containing hydrogels have been observed before (see, e.g., Ader et al., 2010, and Wiegand et al., 2020). Fragai et al. (2013) reported that sedimentation of highly charged proteins typically results in low packing ra- 40 tios in sedimentation, while higher-than-crystalline concentrations can be achieved for proteins with low overall charge. Despite the high overall net negative charge of the nucleosome (−168$e$ for a 167 bp nucleosome), we find packing ratios ∼ 80 % of that in nucleosome crystals. This underscores 45 the crucial contribution of attractive interactions in the nucleosome system, thanks to its separation in negatively charged DNA and positively charged histone proteins.

Previous studies on dense phases of nucleosomes demonstrated formation of highly ordered structures consisting of 50 columns of stacked nucleosomes (Allahverdi et al., 2011; Berezhnoy et al., 2016; Bertin et al., 2007b; Eltsov et al., 2018; Korolev et al., 2012; Leforestier and Livolant, 1997; Livolant et al., 2006; Mangenot et al., 2003a, b). For the isotropic columnar phase, SAXS scattering curves showed 55 three broad scattering peaks corresponding to (i) the average intercolumnar distance, typically at $q^* = \sim 0.065$, with $q^*$ decreasing as the linker DNA increases (Mangenot et al., 2003b; Wang et al., 2021), (ii) the stacking distance between nucleosomes in a column (typically at $q^* = \sim 0.11$), 60 and (iii) the form factor of the column (Livolant et al., 2006). In a highly ordered columnar phase, such as obtained from Mg$^{2+}$-induced precipitation of nucleosome core particles, these peaks appear sharp and well-resolved (Berezhnoy et al., 2016). The scattering curves presented here do not show the 65

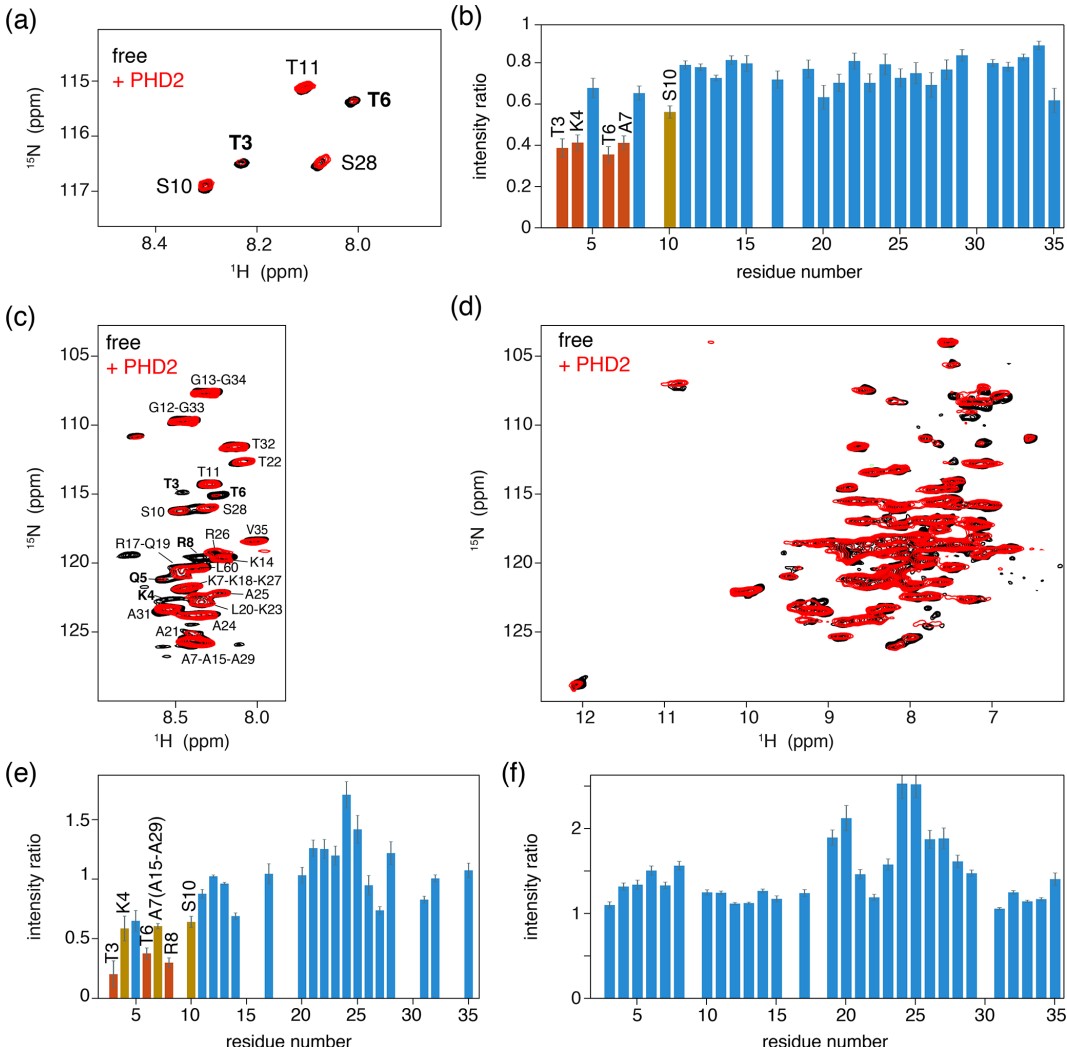

**Figure 4.** PHD2 co-sediments with the nucleosome and has the same effect on the histone H3 tail in the sediments as in solution. **(a)** Comparison of solution NMR spectra of the H3 tail in the nucleosome with and without PHD2, focusing on the Thr/Ser NH region. Molar ratio of PHD2 to H3 tail is 10 : 1. Data recorded in PK100 buffer at 125 mmol/dm³ ionic strength. **(b)** Peak intensity ratio of H3 tail resonances in the nucleosome based on the solution NMR experiments in **(a)**. Addition of 20 equivalents of PHD2 results in large intensity decrease for the N-terminal residues of the tail that comprise the PHD2 binding site. Resonances with peak intensity ratios lower than 1 (2) standard deviations below the 10 %-trimmed average are displayed in orange (red). **(c, d)** J-based **(c)** and CP-based **(d)** spectra of H3-labeled nucleosomes co-sedimented with PHD2, overlayed with the spectra of free H3-labeled nucleosomes. Color coding indicated in the figure. Assignment of H3 tail residues is indicated. Residues with large peak intensity changes are labeled in bold. **(e)** Peak intensity ratio of H3 tail resonances in the nucleosome based on the ssNMR experiments in **(c)**. Co-sedimentation of PHD2 results in large intensity decrease for the N-terminal residues of the tail that comprise the PHD2 binding site. Resonances with peak intensity ratios lower than 1 (2) standard deviations below the 10 %-trimmed average are displayed in orange (red). Reduced intensity ratios for overlapping resonances of A7, A15 and A29 are assumed to represent the effect for A7 based on the observed pattern of changes. **(f)** Peak intensity ratio of H3 tail resonances in the nucleosome between solution NMR spectra recorded in low (PK10 buffer) and high salt (PK100 buffer). Increase in the ionic strength results in higher peak intensities for residues 19–29 while not affecting the peak intensities in the PHD2 binding site.

two most characteristic signals for a columnar arrangement, indicating that our sedimentation approach does not induce such a columnar phase. In retrospect, three factors in our approach may have helped to avoid formation of a strongly ordered sediment. First and foremost, the Mg²⁺ concentration used in our study is way below the minimum required

to precipitate isolated nucleosomes (Berezhnoy et al., 2016; Wang et al., 2021) and, in addition, the use of K⁺ instead of the harder Na⁺ monovalent salt is known to disfavor nucleosome array precipitation (Allahverdi et al., 2011, 2015). Second, we use nucleosomes containing 10 bp of additional linker DNA, adding more net negative charge. Finally, since

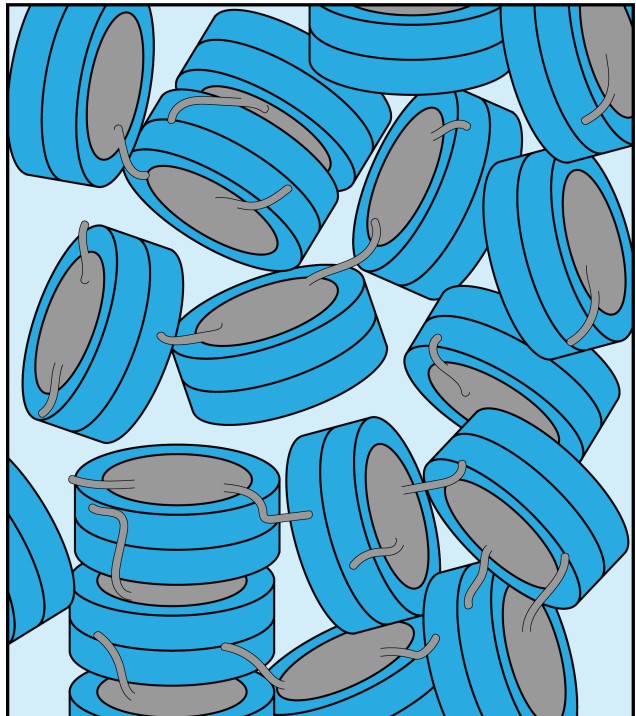

**Figure 5.** Schematic illustration of the packing in the nucleosome sediment obtained by ultracentrifugation. The packing density in the schematic corresponds to our experimental estimate ($\sim 61\%$), while the heterogenous orientation of nucleosomes in the sediment reflects the lack of strong long order in the sample. Nucleosome–nucleosome interactions are predominantly formed by histone tail–DNA interactions.

ultracentrifugation is a relatively fast process, it also impedes the formation of large-scale ordering. To what degree the very fast MAS further impacts the nucleosome packing in the sediment remains to be determined. The presence of a minor component of free bulk water accumulated in the center of the rotor indicates that sample packing increases during MAS due to the much higher centrifugation forces achieved ($\sim 100$-fold). The spinning speeds attained during MAS are so high that the centrifuge effect generates a solvent-based pressure reaching 96 atm near the rotor walls (Elbayed et al., 2005), which will further concentrate nucleosomes locally. Both the appearance and SAXS scattering data of the retrieved samples however indicate that the packing remains mostly disordered.

The dense but disordered nucleosome packing in the sediment suggests that the inter-nucleosome contacts do not stabilize or occlude specific nucleosome surfaces. Indeed, we succeeded here in co-sedimenting a protein that weakly binds the histone tail in the nucleosome, showing that it could effectively compete with the nucleosomal DNA. Surprisingly, binding of PHD2 could only be observed from a peak intensity reduction for the N-terminal residues in the H3 tail that constitute the PHD2 binding motif. This was observed

both in solution and in solid-state NMR experiments. In neither case could a saturation of the binding site be achieved despite the use of a 20-fold molar excess, indicative of a very low binding affinity. The solution NMR experiments indicate that nucleosome binding is ca. 50-fold weaker compared to binding a H3 peptide ($K_D$ 168 vs. 3 µmol/dm$^3$). In the co-sedimentation approach, such weak binding likely blocks quantitative sedimentation of the complex, as dissociated PHD2 molecules will sediment less efficiently and mostly remain in the supernatant.

Surprisingly, no chemical shift changes or signals from the PHD2 bound state of the H3 tail could be observed. Binding of PHD2 can be expected to cause significant loss of flexibility in the H3 tail, as the H3 tail adopts a beta-strand conformation and forms a beta sheet with PHD2 (Mansfield et al., 2011) (see Fig. 6a, b). As the H3 tail is part of the nucleosome, this will broaden the bound-state H3 tail resonances severely in backbone NH-based solution NMR, causing loss of the signal. For ssNMR, reduction of the H3 tail flexibility may push the dynamics into an intermediate regime for which neither scalar- nor in dipolar-based experiments are effective. Inspection of a molecular model of the PHD2–nucleosome complex built using the data-driven docking software HADDOCK highlighted a ridge of positively charged residues on the opposite of the H3 tail-binding site (Fig. 6a, b). To investigate whether H3 tail binding is compatible with simultaneous DNA binding, we allowed for greater flexibility in the H3 tail conformation during docking and imposed ambiguous interaction restraints between the positively charged ridge in PHD2 and the DNA. The resulting models suggest that PHD2 may be able to bind both DNA and H3 tail simultaneously, which would restrain the flexibility of the H3 tail (Fig. 6c). In vitro DNA binding assays of isolated PHD2 did not reveal DNA biding (data not shown), suggesting that H3 tail binding is required to neutralize the negatively charged H3 binding site of PHD2, thus priming the PHD2 positively charged surface for DNA binding. Further experiments will be needed to clarify the molecular details of nucleosome binding by PHD2.

## 5  Conclusion

We examined here the general applicability of the co-sedimentation method for nucleosome NMR studies. The sedimentation procedure robustly produces samples with overall material properties of hydrogels, in which nucleosomes are densely packed in a primarily disordered arrangement. The absence of specific nucleosome–nucleosome interactions renders the method suitable for studying nucleosome dynamics or nucleosome–protein interactions without interference from the higher-order packing of nucleosomes. As a stringent test case we here successfully demonstrated nucleosome binding for a low-affinity histone tail-binding protein. Together, our results give new insights into the sam-

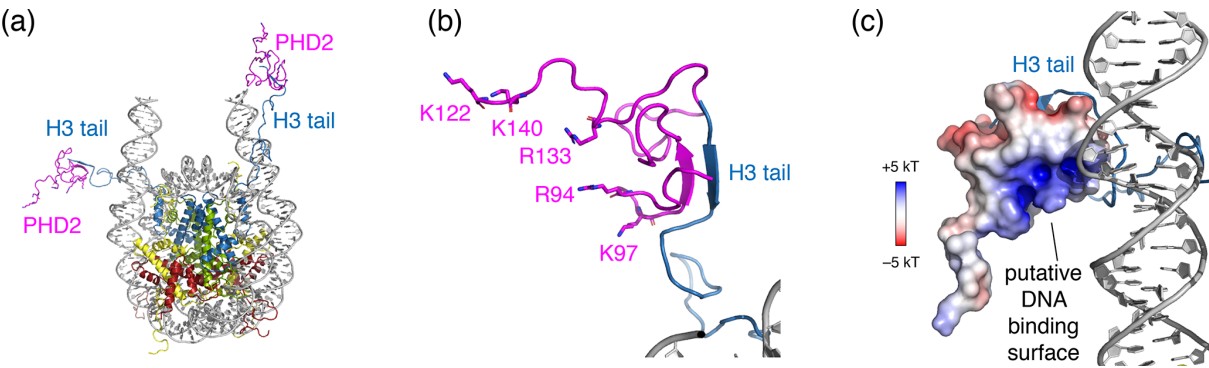

**Figure 6.** Structural model of the PHD2–nucleosome complex derived using HADDOCK. **(a)** Model of the complex based on the H3 tail conformation as seen in the crystal structure (PDB entry 1KX5). The H3 tail residues 1–6 form a beta sheet with PHD2. **(b)** Zoom on the PHD2–H3 tail interaction. Opposite of the H3 tail-binding site, the PHD2 surface features a ridge of positively charged residues, shown as sticks and labeled. **(c)** Model of the complex when enforcing contacts between the positively charge ridge in PHD2 and the DNA, showing contacts between K97 and the nucleosomal DNA. The PHD2 surface is colored by the electrostatic potential calculated by APBS (Jurrus et al., 2018). Note that since the H3 tail is flexible, PHD2 could further reorient, while bound to the H3 tail, to allow more substantial PHD2–DNA contacts. Color coding indicated in the figure.

ple characteristics of nucleosome sediments for ssNMR studies and illustrate the broad applicability of sedimentation-based NMR studies.

**Data availability.** Data are available upon reasonable request.

**Author contributions.** All the authors contributed to designing and planning the experiments. UBlP and SX performed the NMR experiments, MRMH performed SAXS measurements, and YZ made the CHD4-PHD2 protein. UBlP and HvI analyzed the data, and all the authors contributed to finalizing the manuscript.

**Competing interests.** The authors declare that they have no conflict of interest.

**Special issue statement.** This article is part of the special issue "Robert Kaptein Festschrift". It is not associated with a conference.

**Acknowledgements.** We thank Johan van der Zwan and Andrei Gurinov for support and maintenance of the NMR infrastructure. We thank Raymond Schellevis for lab support.

**Financial support.** This research has been supported by the Nederlandse Organisatie voor Wetenschappelijk Onderzoek (grant nos. 723.013.010, 722.016.002, 175.010.2009.002, 718.015.001, and 184.032.207), the National Institutes of Health (grant no. GM125195), and the Horizon 2020 (iNEXT (grant no. 653706)).

**Review statement.** This paper was edited by Jörg Matysik and reviewed by Lars Nordenskiöld, Paul Schanda, and Claudio Luchinat.

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

## Remarks from the typesetter