# Peer review of "Characterization of nucleosome sediments for protein interaction studies by solid-state NMR spectroscopy"

_Magnetic Resonance, 2021_

## Author Response (AR1)

**Response to reviewers**

We thank the reviewers for their comments, that have helped to improve the paper. We have revised the manuscript to address all concerns raised by the reviewers and detail here our point-by-point response.

**Reviewer 1, prof. Luchinat:**

The manuscript by le Paige et al. is an insightful view on solid-state NMR studies of the nucleosome and how to profitably analyse its interactions with partner proteins.

This manuscript reads very well and the SAXS characterization of nucleosome sediments is fascinating. There are some minor presentation flaws that can be very easily addressed, and that I here list in order of appearance in the text:

1) page 2, line 55. The sentence beginning with "Additionally, as opposed" could carry the same references as the previous sentence, plus the already cited Fragai et al. 2013.

  We combined the two sentences to: "This was inspired by seminal studies showing that sedimentation of soluble proteins results in high-quality samples for solid-state NMR with the added benefit that sedimentation is fast, easy-to-use and does not perturb protein folding. (Bertini et al., 2011; Fragai et al., 2013; Gardiennet et al., 2016; Mainz et al., 2015)."

2) To a non-specialist reader "interact in trans" might be quite obscure. Could it be briefly explained?

  We added the explainer by rephrasing to: "Notably, isolated nucleosomes also form tail-mediated interactions with one another – so called *in trans* interactions, …"

3) line 65, "as shown" instead of "shown"

  Corrected.

4) The sentence on page 3, line 67 is particularly important for the development of the manuscript. However, it is not particularly well written, nor emphasized properly. I suggest rewriting. For instance, reversing the order giving first the potential problems, then explaining how the potential problems are ruled out in this work.

  We rephrased the sentence to: "This strong propensity for *in trans* interactions could potentially be favoured in the high concentration samples obtained through the sedimentation approach used in our NMR studies. The formation of high-order structures in which specific nucleosome surfaces are involved in inter-nucleosome interactions could both alter their intrinsic internal dynamics and reduce their availability for protein interactions."

5) Please follow IUPAC recommendations: avoid "molar" and express concentrations in mol/dm3 or mmol/dm3 instead of M or mM.

  Corrected.

6) Still on the emphasis of the different parts, I think that a break at line 74 would be beneficial: "As a test case" would be the beginning of the new paragraph.

We kept the two parts of the study together in a single paragraph to maintain a single concluding paragraph but rephrased this sentence to emphasize the transition: "To assess the impact on the study of nucleosome-protein interactions, we focussed on the second PHD finger of CHD4 as a test case."

7) Line 124: Perhaps references to the rotor filling devices described in 10.1007/s10858-009-9374-3 and 10.1007/s10858-012-9657-y could be appropriate.

We added these by rephrasing to: "Briefly, a custom-made filling device (as described in (Narasimhan et al., 2021), though other similar designs exist, see for example (Bertini et al., 2012; Böckmann et al., 2009; Mandal et al., 2017)) …"

8) Line 136: Processing parameters (apodization function, zero-filling, etc...) are missing. Please provide them.

We added these by rephrasing to: "The J-based NH spectrum was acquired with $t_{1,max}$ 20 ms, $t_{2,max}$ 20 ms and total acquisition time of ~5 h. The FID was apodized with a 30° shifted squared cosine bell function in both dimensions, zero-filled twice in both dimensions, and indirect dimension was extended once by linear prediction before Fourier Transform. The CP-based NH was acquired with $t_{1,max}$ 21 ms, $t_{2,max}$ 20 ms and total acquisition time of ~10 h. The FID was apodized with exponential window function with line broadening of 50 Hz in the direct dimension and a 30° shifted squared cosine bell function in the indirect dimension, both dimensions were zero-filled twice, and the indirect dimension was extended by linear prediction before Fourier Transfrom."

We also added the same info for the solution NMR spectra: "2D 15N-1H TROSY HSQC spectra (t1,max 122 ms, t2,max 67 ms, total acquisition time per spectrum ~2 h). The FID was apodized in both dimensions with a squared cosine bell function and extended once by linear prediction in the indirect dimension before Fourier Transform."

9) Line 147: the scattering angle theta is generally given lowercase

Adapted.

10) Line 175: the amount of residual protein in the supernatant at equilibrium can be calculated by numerical integration over the rotor filling device ( 10.1007/s10858-012-9657-y, 10.1007/s10858-013-9795-x and http://py- enmr.cerm.unifi.it/access/index/sednmr). Do the experimental results match the predictions?

Using the sedNMR webtool, 73% of the nucleosomes are predicted to be in the rotor+cap volume, while we estimate 62% based on the absorbance measurements for sample 4. These values are in reasonable agreement, especially given that the predicted values depend to large extent on the "threshold for immobilization" parameter, which is unknown and is most likely influenced by inter-particle interactions. We used a value of 0.65 in these predictions, lower than the ferritin value of 0.86, which most likely reflects the much

stronger favourable inter-particle interactions for nucleosomes. We thus added: "The efficiency of sedimentation roughly matches that predicted using the sedNMR webtool (Ferella et al. 2013), when considering that favourable inter-particle interactions between nucleosomes may lower the threshold for immobilization."

11) Line 179: the viscous droplet is visible in one of the figures of the "Narasimhan et al. 2021" reference. Why not repeat that photo here?

The droplet is visible in Fig. 1b, we added the reference to the Figure in this sentence and added to the Figure legend: "A droplet of viscous liquid is visible at the top of rotor."

12) Line 184: the concept of "limiting concentration" is rather intuitive, but not typical of standard sedimentation literature, as concentrated solutions are rarely used (10.1016/0003-9861(85)90382-0 and 10.1002/pol.1980.180180909). This concept has been introduced in the equations describing sedimentation in this paper: 10.1039/C1CP22978H.

We rephrased the sentence to: "This resulted in only a ~5–10% increase in final nucleosome concentration, indicating the nucleosome packing is close to maximum at this centrifugation speed."

13) Line 207, add a comma: "We conclude that while"-> "We conclude that, while"

Added.

14) Paragraph 3.4: Co-sedimentation in NMR has been described theoretically in 10.1021/ar300342f and applied in 10.1007/s10858-016-0018-0. Please include these references

We have put these references in the introduction: "This was inspired by seminal studies showing that sedimentation of soluble proteins results in high-quality samples for solid-state NMR, with the added benefit that sedimentation is fast, easy-to-use and does not perturb protein folding (Bertini et al., 2011; Fragai et al., 2013; Gardiennet et al., 2016; Mainz et al., 2015), and can be used to study protein-protein interactions (Bertini et al., 2013; Gardiennet et al., 2016)."

**Reviewer 2, prof. Schanda:**

This is a very interesting article that examines the properties of sedimented samples of nucleosome particles, obtained by ultracentrifugation into ssNMR rotors. The properties of the sample are evaluated by ssNMR and SAXS, and in addition to nucleosome samples, also a complex with a weakly binding protein (PHD2) is studied.

While many groups have been using sedimentation, often without reporting the details of the sample, this study reports their properties, and this will certainly be useful for others that use similar procedures.

The study is rigorously done and I recommend publishing it, after considering the following points.

Page 5, line 138: "weighting factor of the 15N chemical shift differences (in ppm) of 6.51 ". Actually, the weighting factor is 1/6.51=0.154. This is in the range of commonly used value, reviewed here: Williamson, M.P. (2013). Using chemical shift perturbation to characterise ligand binding. Prog. Nucl. Magn. Reson. Spectrosc. *73*, 1–16.

    We corrected the quoted value to 0.154 and added the reference.

The methods indicate that the rotors were closed, but no rubber spacer was used, nor was the cap glued into our rotor. In our hands, the samples can lose water, often excessively, without rubber spacer or gluing. This observation has also been reported in one case: Asami, S., Szekely, K., Schanda, P., Meier, B.H., and Reif, B. (2012). Optimal degree of protonation for 1H detection of aliphatic sites in randomly deuterated proteins as a function of the MAS frequency. J. Biomol. NMR *54*, 155–168; see Figure 11.

The spectra of the 1H water signal change in shape (Fig 2a); there are clearly several components in the first measurement, which disappear after some time. This could be related to loss of bulk water, as described here: Böckmann, A., Gardiennet, C., Verel, R., Hunkeler, A., Loquet, A., Pintacuda, G., Emsley, L., Meier, B.H., and Lesage, A. (2009). Characterization of different water pools in solid-state NMR protein samples. J. Biomol. NMR *45*, 319–327. My interpretation is that:

- There is bulk water, probably in the centre of the rotor, in the beginning – this actually means that the calculated packing density is underestimated, see below.
- The bulk water is lost
- The hydration water is largely (but not totally) retained.

Can the authors comment on this? Have the caps been glued? Probably not, which may explain the losses. For other samples, such water loss can have dramatic consequences on the sample, as we reported for the case of SH3 crystals (reference above, Asami et al).

    Thanks for pointing out the presence of some free bulk water which we had not addressed. There is indeed a minor component of free bulk water that likely evaporates over time (we did not use a rubber spacer and did not glue the caps to be able to retrieve the sample, this now specified in M&M). Despite this water loss the high quality of the spectra over time clearly indicate that the nucleosome remains well-folded and hydrated, even if some hydration water is also lost.

    We added to M&M : ".. by placing the cap without further sealing or inserts."

    We added to main text and rephrased: "A minor component of free bulk water can be observed in the earliest spectra (Böckmann et al, 2009), which disappears over time due to evaporation. We conclude that, while some degree of dehydration occurs, the sediment samples remain overall well hydrated throughout the NMR measurements."

*We rephrased in the discussion: "Nucleosomes remain well-folded and, despite some water loss, remain hydrated in the sediment over the course of several weeks of MAS."*

I am not convinced that the following statement (page 9) is totally true: "Taken together these data demonstrate the nucleosomes in the sediment remain well-folded and hydrated through the measurements without evidence for direct nucleosome-nucleosome contacts" The spectra of day 1 and day 34 are clearly different, and a change in hydration is likely.

*Indeed there are small chemical shift differences in the two spectra. In line 213-214 of the original manuscript we pointed out that the affected residues are close to crystal waters or ions and that thus the changes could be related to changes in hydration. We replaced "demonstrate" to "indicate" in the quoted sentence.*

On page 7, the authors state that the packing density (61%) is lower than in crystals. They state that in crystals the "packing coefficients of ~67% and a solvent content of ~54% ". I did not find out what is meant by "packing coefficient". I assumed that it is the volume occupied by the protein relative to the total volume. But if it was so, then the solvent content should by the complement to the packing coefficient, i.e. 33% in this case. Could the authors clarify this?

*Indeed we did not explain this clearly. We also realized a mistake in the calculation the nucleosome concentration in crystals. This should be ~3.2 mM, based on the unit cell dimensions and Matthew coefficient. We use the packing coefficient as the fraction of total volume occupied by the nucleosome, taken as a solid, completely filled particle with the shape of a disk. In this case, 100% packing would equate to 4.2 mM concentration while the densest packing achievable for disk shaped objects is 90%, which would correspond to 3.8 mM nucleosomes. Nucleosome crystals thus have a packing coefficient of ~76% (3.2/4.2), substantially higher than the sediment. Since the actual nucleosome volume will be smaller than that the enclosing, completely filled disk, the solvent content can be higher than expected from 100% – packing coefficient.*

*We edited the relevant section in the main text to make this more clear and removed the reference to the solvent content in the crystal to avoid confusion. The relevant section in the main text now reads: "The final nucleosome mass in the rotor is estimated to be 1.44 – 1.59 mg, resulting in concentrations in the range of 480 to 530 mg/cm3 or 2.3 to 2.5 mmol/dm3. […] Assuming the nucleosomes to be homogeneously distributed through the volume of the packed rotor and approximating the nucleosome to a disk-shaped object of 420 nm3 (van Vugt et al., 2009), the observed nucleosome concentration corresponds to a packing ratio of ~55-60%. These concentrations and packing ratios of the nucleosome sediment are lower than those found in nucleosome crystals. Based on crystallography parameters from four nucleosome crystal structures (Protein Data Bank (PDB) entries 2PYO, 1KX5, 1AOI and 3LZ0, (Clapier et al., 2008; Davey et al., 2002; Luger et al., 1997; Vasudevan et al., 2010)) we find that the typical concentrations are ~3.2 mmol/dm3, corresponding to a particle packing coefficient of ~76%. These considerations indicate that the sediment is highly dense with a packing ratio close to 80% of that in crystals, suggesting that a significant amount of ordering and nucleosome-nucleosome interactions may occur."*

As eluded to above, it is likely that the packing density in the rotor during MAS is higher than after ultracentrifugation into the rotor. In typical samples, at least crystalline ones, we always find that bulk water accumulates in the centre of the rotor. The water 1H peak in the beginning of the measurement shows an additional component (on the right), which most likely is bulk water, located in the center of the rotor. Thus, I believe that the actual packing density of the species that is detected by NMR is higher than the authors' estimate.

Indeed, the higher centrifugal forces during MAS may increase the packing densities.

We rephrased in the discussion: "To what degree the very fast MAS further impacts the nucleosome packing in the sediment remains to be determined. The presence of a minor component of free bulk water accumulated in the center of the rotor indicates that sample packing increases during MAS due to the much higher centrifugation forces achieved (~100-fold).The spinning speeds attained during MAS are so high that the centrifuge effect generates a solvent-based pressure reaching 96 atm near the rotor walls (Elbayed et al., 2005), which will further concentrate nucleosomes locally. Both the appearance and SAXS scattering data of the retrieved samples however indicate that the packing remains mostly disordered."

Figure 4e: it is surprising that some peaks (res. 20-25) have much higher intensity in the complex (INEPT based). This suggests that these residues become more mobile. Can the authors comment on this?

This effect is due to the lower salt level in the reference spectrum, as mentioned in the text. We added a reference back to Fig 4e for clarity when discussing the salt-dependent peak intensity increase: "Furthermore, addition of salt approximately doubled the peak intensity for many residues in the 19-29 region, indicating increased flexibility for this part of the H3 tail (Fig. 4f) and explaining the higher relative peak intensities for this stretch in Fig. 4e."

**Reviewer 3, prof. Nordenskiold:**

Review of the manuscript "Characterization of nucleosome sediments for protein interaction studies by solid-state NMR spectroscopy" (mr-2021-21) by Ulric B. le Paige et al., by Lars Nordenskiöld and Xiangyan Shi.

le Paige and co-authors presented a thorough discussion about packing nucleosome samples by sedimentation for ssNMR studies. It revealed that the nucleosomes in the ssNMR rotor are highly compacted, well hydrated and stable, meanwhile have sufficient disorder of nucleosome arrangement to allow proteins interacting with nucleosomes. This method allows studying the interactions between nucleosomes and binding factors, as demonstrated by the ssNMR study of a co-sedimented PHD2-nucleosome complex. The contribution is very helpful to practitioners interested in studying nucleosome interactions with NMR.

There are several minor comments.

The authors added 2 mM Mg2+ in the nucleosome solutions prior to sedimentation. What is the reason to choose 2 mM Mg2+? Does it affect the sedimentation efficiency, final

concentration or ssNMR data quality if adding less or adding more Mg2+? See also below on the effect of Mg2+ concentration on order as observed in SAXS X-ray diffraction spectra.

We used 2 mM, far less than required to precipitate mononucleosomes, to avoid pushing the sediment to closely packed state, as mentioned in the Discussion. We did not systematically evaluate the $Mg^{2+}$-dependence of sedimentation efficiency or spectra quality. Judging from results from your lab it seems that both high and low $Mg^{2+}$ conditions result in high quality spectra. We now added the relevant references explicitly in the discussion on the $Mg^{2+}$ level: "First and foremost, the Mg2+ concentration used in our study is way below the minimum required to precipitate isolated nucleosomes (Berezhnoy et al. 2016; Wang et al. 2021), …"

When assessing the final concentration of nucleosomes packed in the ssNMR rotor, have the authors also tried quantifying this by other methods, such as quantifying the 13C or 1H signals using an external standard? It's not uncommon that considerable sample loss can happen during packing, clearing top for caps, closing caps etc, which would lead to overestimating the material quantity in the rotor by checking the supernatant.

As mentioned in Table 1, we explicitly also estimated the material loss upon clearing the rotor top for the cap. We did not use an external standard to determine the nucleosome concentration in the rotor.

In the sedimented PHD2-nucleosome complex sample, 20:1 ratio was used. Does the free PHD2 remain in the supernatant or the sediment? If it is mostly sedimented, is it possible to prevent this by optimizing sedimentation time, or buffer or other conditions? It will be useful to include such information in the manuscript.

The supernatant after sedimentation contained mostly PHD2, suggesting that only the PHD2 molecules that are specifically bound to the nucleosome are sedimented. We made this explicit in the revised version by adding to the results: "The supernatant after sedimentation contained mostly PHD2, suggesting that the specific complex was sedimented while the excess PHD2 remained in solution."

In the introduction, line 60 "mainly via interactions mediated by the histone tails" is simplified, they interact by stacking in columns at various degree of order and the stacking for a condensed system is governed by not only close packing considerations, but the contacts (stacking) between the histone octamer core surfaces are mediated in addition to tails, by charge-charge interactions by charged aa on the two surfaces as well as presence of divalent or higher charge salt counterions that screen electrostatic repulsion.

Thanks for pointing this out. We rephrased the line in the introduction to: "Nucleosomes are well known to interact with each other, mainly via interactions mediated by the histone tails (Garcia-Ramirez et al., 1992; Kan et al., 2007; Schwarz et al., 1996) in addition to other charge-charge interactions."

In many places, the space between a value and the unit is missing, for example, Line106 - 10 mM; Line156, 10 bp; Line 148, 15 s; Line 117, 298 K; and many others.

Corrected.

Line 76, "contacts" -> "contact"

Corrected.

Line 120, "monitored" -> "were monitored"

Corrected.

How were the error bars in Figure 4 derived?

These are derived from the spectral noise level. We added to M&M: "Errors in peak intensities were based on the spectral noise-level."

Regarding SAXS data. "The sediments are devoid of pronounced long-range ordering of nucleosomes". Here they use 2 mM Mg2+. As the authors note later on, the long range order depends highly on the Mg2+ concentration used when precipitating the NCPs. As shown by Berezhnoy et al, the most pronounced order was achieved at 20 mM Mg2+, while there is no long range order at 2 mM.

Indeed, we used low $Mg^{2+}$ with the intent to avoid close packing, which would be induced by high $Mg^{2+}$ as shown by your lab, and here report our assessment on how this approach worked out. As mentioned above, we discuss explicitly the factors that likely contributed to the lack of pronounced ordering, including the choice for low $Mg^{2+}$.

Line 265: "the sediments seem to primarily consist of heterogeneously packed nucleosomes with mean inter-particle distance of 78 nm." Should not this read "7-8 nm"? Line 259 reads: "pronounced peak at q* ~0.08, corresponding to a characteristic distance of ~7-8 nm".

Indeed, this was a Word to PDF conversion problem, now corrected.

Line 348: "we estimate that the length scale of the regular structure in the sediment is ~1520 nm, corresponding to stacks of two to three nucleosomes, without a significant preference in relative orientation between the stacks". How did the authors arrive at this conclusion? Is it based on FWH in SAXS spectra, if so details should be given? The peaks seem too broad and ill-defined for such an analysis.

The line should have read "15-20 nm" (the same PDF conversion problem). This clearly should be taken as a rough estimate of the length scale, based on the FWMH of the first peak. We rephrased the sentence to: "Judging from the large width of the first peak, we roughly estimate that the length scale of the regular structure in the sediment is ~15-20 nm, corresponding to stacks of two to three nucleosomes, at least for sample 2."

It would be good if the authors try to adhere to the recent guidelines for the description of SAXS results described in Acta Crystallogr. D Struct. Biol., 73(Pt 9), 710 (2017). Figure 3c (167 bp nucleosome SAXS solution data) lacks indication of the nucleosome concentration and details of calculation of the form factor profile (nucleosome structure used for calculation, program applied).

We added the requested information to M&M and the Figure caption and added Table 2 describing the SAXS data analysis and included a molecular weight analysis.

For the condensed (precipitated) samples of the 167bp nucleosome, the authors obtained SAXS profiles that are different from data reported for condensed NCPs (nucleosome without linker DNA) and also from the recently published SAXS results for the 177bp nucleosome ( Soft Matter, 14, 9096 (2018); Sci. Rep., 11, 380 (2021)). It would be good

to make some discussion and comparison with the indicated data. Is the difference due to the low Mg2+ concentration in the samples?

We made the discussion a more explicit comparison with previous literature data by rephrasing to: "For the isotropic columnar phase, SAXS scattering curves showed three broad scattering peaks corresponding to the average intercolumnar distance (typically at q* = ~0.065, with q* decreasing as the linker DNA increases (Mangenot et al., 2003; Wang et al. 2021)), the stacking distance between nucleosomes in a column (typically at q* = ~0.11), and the form factor of the column (Livolant et al., 2006). In a highly ordered columnar phase, such as obtained from Mg2+-induced precipitation of nucleosome core particles, these peaks appear sharp and well-resolved (Berezhnoy et al., 2016). The scattering curves presented here do not show the two most characteristic signals for a columnar arrangement, indicating that our sedimentation approach does not induced such columnar phase. In retrospect, three factors in our approach may have helped to avoid formation of a strongly ordered sediment. First and foremost, the Mg2+ concentration used in our study is way below the minimum required to precipitate isolated nucleosomes (Berezhnoy et al. 2016; Wang et al. 2021), and in addition, the use of K+ instead of the harder Na+ monovalent salt is known to disfavour nucleosome array precipitation (Allahverdi et al., 2011, 2015). Second, we use nucleosomes containing 10 bp of additional linker DNA, adding more net negative charge. Finally, since ultracentrifugation is a relatively fast process, it also impedes the formation of large-scale ordering."

**Other changes**

We note that in the initial submission one co-author had mistakenly been omitted and is now included.

We added two more references to work of Rob Kaptein.

We corrected the calculation of the packing coefficient for the sediment, which is 55-60 % instead of 61%.

We added a sentence to specify that a separate sample was made for the SAXS measurement on diluted nucleosomes.

We specified how the nucleosome concentration was determined from absorbance measurements.

We added the approximate temperature during ssNMR MAS experiments.

We corrected the molar ratio PHD2:H3 tail for the solution NMR experiments to 10:1.